# Gut Phageome—An Insight into the Role and Impact of Gut Microbiome and Their Correlation with Mammal Health and Diseases

**DOI:** 10.3390/microorganisms11102454

**Published:** 2023-09-29

**Authors:** Yujie Zhang, Somanshu Sharma, Logan Tom, Yen-Te Liao, Vivian C. H. Wu

**Affiliations:** Produce Safety and Microbiology Research Unit, U.S. Department of Agriculture, Agricultural Research Service, Western Regional Research Center, Albany, CA 94710, USA

**Keywords:** gut phageome, phage–bacterial interaction, mammal health and diseases, phage therapy

## Abstract

The gut microbiota, including bacteria, archaea, fungi, and viruses, compose a diverse mammalian gut environment and are highly associated with host health. Bacteriophages, the viruses that infect bacteria, are the primary members of the gastrointestinal virome, known as the phageome. However, our knowledge regarding the gut phageome remains poorly understood. In this review, the critical role of the gut phageome and its correlation with mammalian health were summarized. First, an overall profile of phages across the gastrointestinal tract and their dynamic roles in shaping the surrounding microorganisms was elucidated. Further, the impacts of the gut phageome on gastrointestinal fitness and the bacterial community were highlighted, together with the influence of diets on the gut phageome composition. Additionally, new reports on the role of the gut phageome in the association of mammalian health and diseases were reviewed. Finally, a comprehensive update regarding the advanced phage benchwork and contributions of phage-based therapy to prevent/treat mammalian diseases was provided. This study provides insights into the role and impact of the gut phagenome in gut environments closely related to mammal health and diseases. The findings provoke the potential applications of phage-based diagnosis and therapy in clinical and agricultural fields. Future research is needed to uncover the underlying mechanism of phage–bacterial interactions in gut environments and explore the maintenance of mammalian health via phage-regulated gut microbiota.

## 1. Introduction

The mammalian gastrointestinal (GI) tract has a diverse microbial environment containing bacteria, viruses, yeast, and fungi [1,2]. As one of the most populated microbial environments, the gut microbiota plays a critical symbiotic role in mammal hosts by encoding many genes that metabolize nutrients and extract enough energy from food [3]. Several bacterial phyla, such as *Proteobacteria*, *Firmicutes*, *Actinobacteria*, and *Bacteroide*, compose the majority of the gut microbiota and contribute to the metabolism and health of mammalian animals [4,5,6]. In addition, depending on lifestyle, food, environment, and geographical divergences, the gut microbiome of different individuals shows varying populations of microorganisms. For example, people from rural areas have a high population of *Proteobacteria* and *Spirochaetes*, while urban communities contain a great enrichment of *Firmicutes* and *Bacteroides* [7,8,9,10]. American communities contain a high abundance of *Firmicutes* in their gut microbiome, whereas both Japanese and Korean communities showed a high population of *Actinobacteria* [11].

The diversity and variability of the gut microbiome also play crucial roles in host health. Research has indicated that various gut microbes have a fundamental role in the mammalian immune system [6]. On the one hand, the gut microbiota primes immune responses by lightly activating the host’s immune system enough to prepare the body against infection but does not overstimulate the body [6,12]. For instance, one of the major fungal species—*Candida albicans*—was found to significantly induce T helper 17 cell responses and regulate mucosal immunity in multiple sites along the human gut [13]. On the other hand, much research has found that changes in the gut microbiota were also highly related to the diseases of the mammalian hosts [14]. For example, asthma was correlated with lower-than-normal bacterial levels of *Faecalibacterium* and *Lachnospira* in children [15]. Compared to healthy individuals, colorectal cancers (CRCs) are associated with higher bacterial diversity and reduced temporal stability; in particular, higher levels of *Bacteroides* and lower levels of *Firmicutes* were observed in patients’ gut [16]. These studies established the association of the gut microbiota, primarily bacteria, with host health. 

Bacteriophages (phages), the prokaryotic viruses infecting bacteria, have predominately driven the bacterial population through the phage–bacterial ecology and co-evolution in different natural environments [17,18,19]. Specifically, phages infect bacteria through a strictly lytic or lysogenic cycle [20]. Lytic phages (or virulent phages) inject their phage genomes into bacterial hosts and use hosts’ machinery to replicate phage genomes and produce viral progeny, causing immediate bacterial lysis to release phage particles. Lysogenic phages (or temperate phages) integrate their genomes into the bacterial chromosomes as prophages or present as free plasmids without causing instant bacterial lysis, thereby being able to replicate with the bacterial cells in the lysogenic cycle. Certain external stresses, such as antibiotics commonly used in clinical treatment, can induce prophages from bacterial genomes to produce infectious phage particles. In the gut environment, phages were the primary members of the gastrointestinal virome, also known as the gut phageome [21]. Researchers display diverse putative mechanisms to characterize phage–mammal-host interactions and continually expand on different phage families. Furthermore, across mammalian guts, there is a common reservoir of phages found in more than 50% of healthy human individuals worldwide, suggesting the existence of the healthy gut phageome (HGP) [22,23,24]. In contrast, changes in the gut virome, particularly dsDNA phages, were positively correlated with bacterial dysbiosis and increased GI epithelial permeability in Gulf War illness (GWI) patients [25]. The findings provide new insights into the critical role of gut bacteriophage in mammalian health. Therefore, the question of what degree and how the gut phageome contributes to the dynamic of gut microbiota and mammal health is important to address. 

The field of gut microbiota has continued to expand rapidly over the years; however, our knowledge regarding the gut virome, especially the gut phageome, remains poorly understood and requires further elucidation. Thus, this review aims to provide new insights into the critical role of bacteriophages in the gut microbiome and their correlation with mammalian health. In this review, we first summarized recent studies to provide an overall profile of phages across the gastrointestinal tract and their dynamic roles in shaping the surrounding microorganisms. Next, we highlighted the positive and negative impacts of the gut phageome on gastrointestinal fitness and the bacterial community, as well as the influence of diets on the composition of the gut phageome. Furthermore, we reviewed new findings on the vital role of the gut phageome in regulating and maintaining mammalian health. Deviations or disruptions in the ecology of the gut phageome have been implicated in the emergence of several diseases and metabolic disorders, such as ulcerative colitis (UC) and inflammatory bowel disease (IBD). Finally, we provided a comprehensive update on the application of phage benchwork on new advances and contributions of phage-based therapy to prevent/treat mammalian diseases. Advanced technologies, including viral metagenomics, phage-based CRISPR gene editing of gut microbiota, and fecal viral transplantation, used in clinical trials and agriculture, were discussed in this review. This review gathers multiple research findings regarding the gut phageome and its implementation in clinical and agricultural fields to focus on creating a consensus perspective on how phages benefit mammalian health. It is also the first review to highlight the gut phageome and its interactions with mammalian health and diseases as well as phage-based therapy against mammalian diseases.

## 2. Phages in the Mammalian Gut

The GI tract is a heterogeneous ecosystem that contains diverse organisms and distinguished sub-environments with distinct microbial communities structured to fit their role in the digestive system. The phage populations were found across virtually every organ and tissue of the GI tract, suggesting an active and important role in the mammalian gut microbiota [23]. To investigate the spatial distribution of the phageome in the gut, Bao et al. fed 8 log PFU/mL of lytic phage PA13076 or lysogenic phage BP96115 to mouse models for 31 days to track phages; they observed that the two phages gradually increased toward the end of the GI tract, with the highest phage titer (close to 6.5 log PFU/g) present in feces and the lowest in the stomach and duodenum [26]. A similar phenomenon was observed in humans that the unique environment of each GI site created its microbial niche specificity. There are generally more phages toward the end of the GI tract, with high phage titers found in the gut colon lumen when feces are present in the colon [27]. Despite the information regarding the human gut phageome from different GI sites being very limited due to challenges in sampling, a few studies have provided the phage profile within the different gut locations of other mammal animals, such as rhesus macaques [28]. For example, a great abundance of phages belonging to the *Caudovirales* order (mostly the *Myoviridae*, *Podoviridae*, and *Siphoviridae* families) and the *Microviridae* family was detected in different GI sites (terminal ileum, proximal colon, distal colon, and rectum) of the rhesus macaque [28]. Moreover, the researchers also indicated that the phageome from ileum samples was distinct from that from the colon and rectum, demonstrating various types and abundance of phages within different gut locations based on the structural characteristics in the gut. Overall, the phageome profile of mammalian animals is ultimately unique and diverse across the GI tract.

Moreover, the diversity and plasticity of the phage genome can also improve phage adaptability to GI sites and form niches in the mammalian gut environment. In a gut-on-a-chip mucosal environment, evolved phage populations, particularly those with a genetic mutation on phage capsid protein, could cause the altered glycan-binding phenotype, showing a competitive fitness advantage over ancestral phages [29]. Specifically, phages with mutated phage capsid protein Hoc facilitate phage adherence to mucus via binding to human fucosylated mucin glycans and are further localized toward bacterial hosts [29]. Additionally, compared to the control group (phage–bacterial co-evolution without a mammalian mucosal environment), increasing phage variations with distinct gene mutational profiles were observed in the group with a mammalian mucosal environment. The evolution of these phages in response to the dynamic mammalian gut environment demonstrates a trans-domain evolutionary process along the phage–mammalian axis [29]. Similarly, the tail fiber genes encoding proteins in some phages can bind to human heparan-sulfated proteoglycans and later place phages near the epithelial cell surface, facilitating contact with bacterial hosts and infection [30]. For example, a tail fiber protein coded by phage ES17 allows the phage to bind to mammalian polysaccharides, enhancing the phage antimicrobial activity in a mucin-rich environment. Moreover, the phage could adhere to the mucus layer and the intestinal epithelial cell (IEC) surface by binding to heparan sulfate proteoglycans (HSPGs) via this unique tail fiber protein and target its bacterial hosts adjacent to the area in the intestinal microenvironment [30].

## 3. The Influence of Diet on the Changes of Gut Phageome

Phageomes are established prenatally with the transmission of microorganisms maternally. Co-twins and their mothers exhibited a significantly higher degree of virome similarity than unrelated individuals [31]. Infancy is a crucial period for phageome development; malnourishment in infancy greatly disrupts a healthy phageome, while malnourished youth contains a phageome similar to individuals with growth stunts and various gut illnesses [31,32]. With proper diet, infants establish base levels of bacteria, lysogens, and free phages in the gut and increase populations of *Caudovirales* phages, which will later become a hallmark of the mammalian gut phageome. 

During the life growth of mammalian animals, diets have a considerable influence on shaping the gut phageome [23,33]. Food macromolecules, such as proteins and carbohydrates, induce shifts in the gut microbiota. Multiple studies have shown changes in diets linked with changes in the gut phageome over the growth of mammalian animals [23]. Several foods have been reported to directly impact the profile of the human gut phagenome, subsequently affecting the gut bacteriome. Garmaeva et al. studied phages in the guts of healthy adults on a gluten-containing diet (control) or a gluten-free diet (GFD). In contrast to the control group, crAss-like phages, *Microviridae*, and *Podoviridae* phages were decreased in the gut phageome from GFD individuals. In addition, Bray–Curtis distances between the two groups were altered at various time points, revealing changes in the gut phageome [34]. Certain foods, like fresh meat and fermented foods, are rich in phages, so the diets concerning these foods may contribute to a wide range of gut phageomes [35]. In addition to the individual gut phageome, diets from geographical communities and ethnic cultures also play significant roles among different populations. Zuo et al. studied the gut phage DNA virome in 930 healthy individuals in Hong Kong and Yunnan, China, who spanned various ethnicities and residencies, and geography played the most significant role in shaping both an individual’s gut virome and bacteriome [36]. It is not clear whether the virome affected the bacteriome or the other way around; however, certain foods, such as barley, buttermilk tea, and Pu’er tea, have been ethnically established in these regions and contributed to virome differences. 

Although the information regarding the impact of diets on the human gut phageome is limited, several studies have revealed the relationship in other mammalian animals. For example, the gut viruses belonging to *Caudoviruses* were predominant in mice fed with normal chow; however, the mice fed with a high-fat diet showed an increased relative abundance of *Microviridae* in the gut virome [37]. Hallowell et al. (2021) studied how different diets influence obese pigs, and obese pigs fed with ad libitum diets saw an increase in various *Myoviridae* phages [38]. In addition, researchers reported a rapid and significant decrease in phages targeting *Streptococcus* spp., along with *Streptococcus* spp., in obese pigs. To understand the mechanistic details of alternative phage profiles by diets, Kim and Bae (2018) studied the effects of different diets on murine gut microbiota, specifically the populations of phages with different lifecycles. Their results revealed that more lytic phages were observed in mice with low fat, high plant polysaccharide (LFPP) diets, while lysogenic phages were much more prevalent in mice with high fat, high sucrose (HFHS) diets [39].

Overall, these findings provided evidence about the association between diets and alterations in the gut microbiota of mammalian animals in general, particularly in the phageome. It is important to understand how various foods affect the profile of the host gut phagenome and the further effect on the entire gut microbiota.

## 4. How Do Phages Shape the Gut Microbiota?

In certain microbiota, phages do not simply live with and infect bacteria but play dynamic roles in shaping their surroundings. The gut phageome lyses bacteria depending on the density of bacterial species; this drives microbial diversity and evolution and stabilizes the microbial population [40,41]. On the one hand, the phageome is a crucial part of shaping the gut microbiota and the health of the host body [21,42,43]. Stern et al. reported specific phages or taxa of phages in the human population correlated with human health [42]. Researchers identified 991 phages from the gut microbiota of 124 individuals using sequences found in bacterial CRISPR genes, and 78% of these phages were present in two or more healthy individuals. Also, some lysogenic phages can transport important genes, such as those involved in anaerobic respiration and macromolecule biosynthesis, between different bacterial cells through transduction, contributing to the gut microbiota’s ecological role. Attai et al. reported that the addition of allochthonous phages affected the metagenomics of the bacterial diversity and composition in an in vitro bioreactor model of the human gut [44]. Their findings demonstrate that phages play active roles in the gut microbiota and affect multiple bacterial phyla, including *Bacteroidetes* and *Firmicutes*. On the other hand, phages can also disrupt gut health and cause damage to the GI tract. Duerkop et al. reported that when lysogenic phages ΦV1/7 infected human-specific *Enterococcus faecalis* clinical strain V583, the prophages appeared to give *E. faecalis* a competitive pathogenic advantage in both in vitro and in vivo models, increasing intestinal inflammation [45]. Lourenço et al. observed that exogenous phage CLB_P3 did not infect bacteria but instead increased the biofilm formation in enteroaggregative *E. coli* strain 55989, which resulted in more disease in mice models [46].

Therefore, the phage–bacterial dynamic within the GI environment is important to investigate due to its impact on the shape of the gut microbes and its close relationship with host health (Figure 1). Scanlan et al. observed lytic phages playing an ecological role in the gut of phage-cocktail-treated mice [47]. After phage treatment, the gut bacteria exhibited a source-sink ecological dynamic where bacteria moved between different gut areas based on overpopulation and nutrient availability. Phages and bacteria can illustrate the Red Queen dynamics. Specifically, there is a continual co-evolution between the host and phage to defend and counter-defend, each species running to keep up with the other. In addition, phages can display kill-the-winner dynamics to lyse susceptible cells and prevent bacterial dominance. They also exhibit piggyback-the-winner dynamics, where lysogenic phages integrate with their host and co-exist [23,27]. The piggyback-the-winner dynamic is predominant in the absence of pathogens and plays a crucial role in genetic exchange; prophages have shown up to 5% of the functions in the human gut microbiota, such as nutrient cycling and population stability [48]. Although few bacterial species are able to successfully escape phage predation, a more realistic perspective of the interactions between phages and bacteria is mutualistic. Phages have not been shown to eliminate gut bacteria but constantly control bacterial density and distribution [49,50]. Despite the risk of lysis, the gut microbiota benefits from phages in the GI tract through lysogeny and regulation of colonization, and phages have formed an important ecological niche with GI microbes. Based on these findings, a broad and evident regulation network among phages, the gut microbiome, and mammalian health is better known; however, detailed information regarding how phages are involved in gut disease and therapy is still needed.

## 5. Phagenome, Health, and Disease

In addition to the crucial part of shaping the gut microbiota, the phageome has been further confirmed to play a vital role in regulating and maintaining host health. For example, high *Caudovirales* and *Siphoviridae* levels in the human gut microbiome were associated with better performance in brain executive function and verbal memory [52]. Moreover, the gut-residing phageome can also boost the host immune system to combat pathogens and maintain a healthy gut microbiome. Research by Barr et al. showcased that phages with immunoglobulin-like domains in their capsids were able to bind to mucin in the mucosal surfaces of all animal guts, resulting in antibacterial protection against pathogens [53]. Yet another study indicated that phages adsorbed into the bacteria within the host mice guts and were able to stimulate macrophages of the mouse immune system and reduce cytotoxic damage caused by methicillin-resistant *S. aureus* (MRSA) [54].

Deviations or disruptions in the gut microbiota ecology have been implicated in the emergence of several diseases and metabolic disorders (Table 1) [55,56,57,58,59,60,61,62,63,64]. For example, in comparison to healthy controls, patients with ulcerative colitis (UC) showed an expansion of mucosa phages as well as a decrease in the diversity and richness of phages from the *Caudovirales*, whereas there was also an increase in *Escherichia* phage and *Enterobacteria* phage populations in the guts of UC patients [65]. Another study by de Jonge et al. investigated the connection between metabolic syndrome (MetS) and changes to the gut phagenome. The researchers found that in a sample of 196 clinical patients, those with MetS had a significantly higher abundance of phages infecting *Streptococcaceae* and *Bacteroidaceae* and a significantly lower abundance of phages infecting *Bifidobacteriaceae*. An overall lower phage diversity and richness was also observed in the patients compared to healthy controls. Extensive research has also been conducted to examine the changes in the gut virome in cases of Crohn’s disease (CD). One such study examined changes in the gut prokaryotic virome in Japanese patients with CD [66]. The researchers determined that the fecal samples from CD patients exhibited significantly higher levels of crAssphage compared to control fecal samples. Additionally, some viruses were unique to only control or CD samples: *Lactococcus*, *Enterococcus*, and *Lactobacillus* phages were found exclusively in the CD fecal samples, and *Xanthomonas* and *Escherichia* phages were only present in control samples. Other human diseases, such as diabetes, were also shown to correlate with the gut phageome. Tetz et al. noted that an increased *E. coli* phages/*E. coli* ratio, resulting from the prophage induction, was responsible for the emergence of type I diabetes [67].

The connection between phageome and diseases is also confirmed in animals. For example, colitis in rodents was associated with the changes in phage communities in the gut, most notably a decrease in the abundance of *Clostridiales* phages [68]. Additionally, treating germ-free mice with *Lactobacillus*, *Escherichia*, and *Bacteroides* bacteriophages led to significant inflammation in the intestine and even the cause of colitis [69]. Moreover, the gut phagenome can also be associated with infection from foreign viruses. Research by Bao et al. investigated alterations to the gut virome in mice infected by *Coxsackievirus* B3. The results revealed that the foreign B3 virus infection caused a significant increase in *Caudovirales* and a significant decrease in *Microviridae* in the gut virome.

**Table 1 microorganisms-11-02454-t001:** The correlation between human diseases and the gut phageome.

Diseases	Host	Key Findings Regarding Gut Phageome	References
Systemic lupus erythematosus (SLE)	Humans	Patients with SLE contained significant levels of *Siphoviridae*, *Microviridae*, and *crAss-like* viruses, while healthy control primarily exhibited *Siphoviridae* and *Myoviridae*.	[70]
	Humans	Patients with SLE exhibited higher proportions of bacteriophages than the healthy controls. Particularly SLE patients showcased a higher abundance of viral families *Demerecviridae* and *Phycodnaviridae* in the gut virome than the controls.	[71]
Osteoarthritis (OA)	Humans	A total of 122 viral operational taxonomic units (vOTUs) were identified as being higher in OA patients than the healthy controls. These vOTUs contained 7 known viral families, including *Siphoviridae* viruses. On the other hand, approximately 505 vOTUs were found to be depleted in OA patients compared to the healthy controls. These vOTUs consisted of 10 viral families, including *Siphoviridae*, *Myoviridae*, and *Microviridae* viruses.	[72]
Atopic dermatitis (AD)	Humans (Children)	Viruses from the *Microviridae*, *Myoviridae*, *Mimiviridae*, and *Siphoviridae* families were at significantly different levels between patients with AD and healthy controls.	[73]
Crohn’s disease (CD)	Humans	Patients with CD exhibited increased levels of *Faecali* phages and *Escherichia* phages compared to healthy controls. Healthy subjects showcased higher levels of some prokaryotic viruses compared to CD patients.	[74]
	Humans	Patients with CD had higher levels of *crAssphage* than the healthy controls. *Lactococcus*, *Enterococcus*, and *Lactobacillus* phages were exclusive to CD patients, while *Xanthomonas* and *Escherichia* phages were exclusive to control patients.	[66]
	Humans	Patients with CD had a significant expansion of *Caudovirales* bacteriophages; temperate bacteriophages dominate the gut virome.	[75]
Sepsis-induced cardiomyopathy (SICM)	Humans	SICM patients showed higher levels of *Cronobacter* and *Cronobacter* phages than the subjects with sepsis-uninduced cardiomyopathy.	[76]
Metabolic syndrome (MetS)	Humans	MetS gut viromes showed increased levels of phages infecting *Streptococcaceae* and *Bacteroidaceae* and decreased levels of phages infecting *Bifidobacteriaceae*.	[77]
Human immunodeficiency virus (HIV)	Humans	HIV-infected patients showcased an increased abundance of lysogenic phages compared to healthy controls. This increased level of lysogenic phages remained even after treatment with integrase strand transfer inhibitors (INSTIs).	[78]
Liver cirrhosis	Humans	Patients with liver cirrhosis showcased prevalent levels in *Siphoviridae*, *Podoviridae*, and *Myoviridae*.	[79]
Necrotizing enterocolitis (NEC)	Humans (Preterm infants)	A total of 137 viral contigs were found to appear 0–10 days before NEC onset. These contigs belonged to many viral families, including *Myoviridae*, *Podoviridae*, and *Siphoviridae*.	[80]
Critical congenital heart disease (CCHD)	Humans (Neonates)	CCHD patients had an increase in α-diversity of gut virome compared to the healthy control (HC). Also, prophages in genomes of *Proteobacteria* and *Bacteroidetes* were elevated when there was a decreased proportion of *Actinobacteria* with prophages in the CCHD group compared to HC.	[81]
Colorectal cancer	Humans	Compared to healthy controls, patients with colorectal cancer exhibited depleted levels of *Siphoviridae* but increased levels of *Microviridae*, *Autographiviridae*, and *Gratiaviridae*.	[82]
	Humans	The viral family *Herelleviridae* was found to be depleted in patients with colorectal cancer.	[83]
	Humans	The diversity of the gut bacteriophage community was significantly increased in colorectal cancer patients.	[84]
Polycystic ovary syndrome (PCOS)	Humans (Females)	The *Bacteroidaceae* phages were predominant in the vOTUs of PCOS patients, while control vOTUs exhibited viruses mainly from *Oscillospiraceae* and *Prevotellaceae* phages.	[85]
Type I diabetes (T1D)	Humans	There were 25 phages identified at statistically different levels in T1D patients compared to the healthy controls. Of these, six phages (uvig_37554, uvig_280596, uvig_296393, uvig_436746, uvig_514207, uvig_557689) changed in abundance, with increasing levels of albuminuria.	[86]
	Humans	T1D patients had an increase in the *E. coli* phage/*E. coli* ratio due to prophage induction.	[67]
Type II diabetes (T2D)	Humans	In T2D patients, the relative numbers of the *Myoviridae*, *Podoviridae*, *Siphoviridae*, and unclassified_*Caudovirales* families increased significantly.	[87]
	Humans	T2D patients showed an increase in the abundance of phages specific to *Enterobacteriaceae* hosts.	[88]
Ulcerative colitis (UC)	Humans	Patients with ulcerative colitis showed a decrease in the diversity and richness of *Caudovirales* phages and an increase in the abundance of *Caudovirales* phages and *Enterobacteria* phages in particular.	[65]
	Humans	There was an over 10-fold higher concentration of virus-like particles (VLPs) from *Siphoviridae*, *Myoviridae*, and *Podoviridae* morphotypes in patients with UC.	[89]
Acquired immunodeficiency syndrome (AIDS)	Humans	An increase in adenoviruses and viruses from the *Anelloviridae* was observed in the viromes of AIDS patients.	[90]
Stunting	Humans (Children)	Gut viromes of stunted children showcased lower phage diversity and a decrease in temperate phages.	[91]
Cardiovascular diseases (CVDs)	Humans	Elevated levels of *Propionibacterium* phages, *Pseudomonas* phages, and *Rhizobium* phages were associated with CVDs.	[92]
Hypertension	Humans	Viruses could have a superior resolution and discrimination power than bacteria for differentiation of healthy samples and pre-hypertension samples, as well as hypertension samples. Viruses such as *Streptococcus* virus phiAbc2, *Cronobacter* phage CR3, and *C. medinalis* granulovirus were associated with hypertension.	[93]

These studies thus solidify the link between the gut phageome and the emergence of various diseases. Studies like these showcase the value of using gut phageome as biomarkers of several metabolic and gastrointestinal diseases in the future.

## 6. Phage-Based Application to Control/Prevent Host Diseases

Diverse phages play critical roles in mammals’ health and diseases. In contrast, some diseases, such as the infection of foreign viruses, also have been implicated in disrupting the gut microbiota ecology and affecting the structure of the gut phageome. Thus, the application of phage research can help provide a useful framework for better diagnosing and controlling pathogens through the regulation of the gut microbiome (Figure 2).

### 6.1. Viral Metagenomics

Viral metagenomics has been a promising approach for various viral research [94,95,96]. The viral metagenomics technique involves collecting viruses from a community of organisms and characterizing their genomic features using bioinformatic tools. The application of viral metagenomics in the context of the gut microbiome can help researchers determine the causative factors of a variety of gut-related diseases as well as understand the gut virome in greater detail. Fernandes et al. showed that more Caudovirales phages were found in children with CD than those with UC [97]. Additionally, compared to healthy controls, the children with CD had a lower richness of phages from *Microviridae*. These findings indicate that viral metagenomics is a very useful tool in understanding the makeup and composition of the gut microbiome and how changes in this composition can lead to different diseases. Viral metagenomics has also helped expand the study of the pathogenesis of autoimmune diseases and gastrointestinal disorders. Particularly, classifying gut viromes can help determine what kinds of viruses may be responsible for triggering the immune system to target the host’s own body. A study by Kim et al. found that 129 viruses were statistically significantly different between the samples from children with islet autoimmunity and healthy children [98]. Most importantly, five enterovirus A species were detected at significantly higher levels in the islet autoimmune samples than in controls. These results highlight a potential causal link between enterovirus A species and the emergence of islet autoimmunity in children. Furthermore, the results underscore the effectiveness of gut viral metagenomics as a laboratory technique to analyze the pathogenicity of diseases. In addition, viral metagenomics has assisted in solving many mysteries about gastrointestinal disorders. While several pathogenic substances and metabolic conditions could lead to diarrhea, there was a rise in unexplained severe diarrhea in children from Turkey in 2015. After investigating the possible etiological agents, Altay et al. discovered that *bufavirus* DNA was present in about 1.4% of the diarrhea samples but none in the samples from healthy children [99]. Additionally, the samples containing the virus of *bufavirus* genotype 3 had more severe diarrhea than other samples. Another study by Yinda et al. showcased that viruses were abundant from the families of *Adenoviridae*, *Astroviridae*, *Caliciviridae*, *Picornaviridae*, and *Reoviridae* in fecal samples from patients with gastroenteritis symptoms [100]. Additionally, the authors indicated that the viruses isolated from patients’ fecal samples, including *orthoreoviruses*, *picobirnaviruses*, and *smacoviruses*, shared a genetic similarity with viruses isolated from fecal samples of bats and other animals. Thus, these results highlight not only the importance of viral metagenomics in determining the cause of diseases such as diarrhea but also underscore how genetic tests can potentially be used to trace back the source of these etiological viruses to other animal hosts.

The field of viral metagenomics is not only limited to humans but is also applicable to animals. Understanding the components of the gut viromes of farm animals through viral metagenomics can help identify pathogens that could have disastrous consequences for food production. Namonyo et al. used rumen viral metagenomics to show that bacteriophages are the majority of the rumen viruses and play a key role in keeping the population count of bacteria in the microbiome of the rumen under the carrying capacity, which is vital for the health of sheep and goats [101]. Viral metagenomics is also useful for identifying causative agents from many animal diseases that could negatively affect food production [102,103]. For example, Wüthrich et al. identified six potential viral causes of non-suppurative encephalitis in cattle: PIV-5, BoAstV-CH13/NeuroS1, bPyV-2 SF, OvHV-2, BHV-6, and BoRV-CH15, through viral metagenomics sequencing [104]. Another study conducted by Ng et al. found potential causative agents were highly associated with agents of bovine respiratory disease (BRD) using viral metagenomic, including bovine adenovirus 3, bovine adeno-associated virus, bovine influenza D virus, bovine parvovirus 2, bovine herpesvirus 6, bovine rhinitis A virus, and bovine rhinitis B virus [105]. 

As viral metagenomics continues to be utilized for gut virome research, several challenges remain to overcome in the future. For example, most viral samples indicated the presence of unknown viruses, which held significantly little sequence similarity to viromes in the GenBank database. It also poses difficulty in linking the phages with their bacterial hosts and understanding their interactions and further impacts on mammalian diseases. In addition, compared to the microbial metagenomic data, current bioinformatic tools regarding viral metagenomic data are limited. Better standardization of viral metagenomic analysis is urgently needed to help lead to a broader understanding and characterization of the gut phagenome. 

### 6.2. Phage Therapy

Phage therapy is an emerging field that uses bacteriophages to selectively target and eliminate pathogenic bacteria. This novel approach also has its advantage over traditional antibiotics, which are often not specific to harmful bacteria and can thus have the side effect of eliminating beneficial gut bacteria as well. As described earlier, dysbiosis in the gut, through the imbalance of bacterial species, for example, can lead to a variety of health problems. Thus, the selective elimination of these harmful bacteria through phage therapy can be a breakthrough in treating several health disorders. The specificity and efficiency of phage-based therapy can potentially make this technology the dominant medical procedure for curing gastrointestinal diseases. 

Phage therapy has not yet been approved for human use in the United States. Only a few cases have been approved for experimental phage use by the US Food and Drug Administration (FDA) in a single-use Investigational New Drug [106,107,108]. However, research in the phage therapy field is still persistent, and the medical potential of phage therapy as an antibacterial treatment is being studied globally. While phage-based therapy development is still in its early stages for clinical applications, animal models have been used to explore the potential clinical applications of this technology. Contemporary phage therapy has relied on lytic phages and usually involves multiple phages to create “a phage cocktail” for greater efficacy [109,110,111]. Work by Maura et al. tested the efficacy of a cocktail containing three lytic bacteriophages (CLB_P1, CLB_P2, and CLB_P3) in eliminating the enteroaggregative *E. coli* (EAEC) O104:H4 55989Str strain using mouse models [112]. The results showed that after 24 h, the phage treatment led to significantly lower concentrations of 55989Str in the ileum of the mice and slightly lower concentrations in the fecal matter. Thus, the phage efficacy and potential application in eliminating pathogenic bacteria in the gut is evident. However, the regrowth of 55989Str after three days of the treatment reveals the limitations of contemporary phage therapy. Thus, more research is needed to resolve the obstacle and design long-lasting treatments against bacterial pathogens. However, phage therapy has the potential as a supplemental treatment to aid other therapeutic procedures. One of the primary medical areas the phage-based therapy has been tested on is cancer. Particularly, phages have been explored as supplements to conventional cancer therapies, like chemotherapy. Research conducted by Zheng et al. studied mouse models with colorectal cancer to test the ability of phages to aid chemotherapy in treating cancer [113]. The bacteria *Fusobacterium nucleatum* likely promote tumors along the gastrointestinal tract, leading to colorectal cancer. Thus, the researchers used irinotecan-loaded dextran nanoparticles covalently linked to azide-modified phages to selectively eliminate harmful *F. nucleatum* in the gut. The results revealed that this phage-based treatment led to significantly more successful first-line chemotherapy treatments in mice compared to the controls. Additionally, the authors repeated experiments in piglets, and there were no significant changes in liver and renal functions. Their findings suggest that phage-based therapy can be a potential supplemental treatment for colorectal cancer with little-to-no side effects. 

Phage-based therapy has also been studied in the agriculture field with great detail. Extensive research has emphasized the use and efficacy of phage therapy against pathogens related to animal disease in the agricultural industries. For example, phage-based application of reducing *Staphylococcus aureus* infection in livestock and other food-producing animals has been studied by researchers. In a study by Gill et al., *S. aureus* bacteriophage K was used to treat mastitis in cattle caused by *S. aureus* infection and resulted in a cure rate of 16.7% in the cattle, while no cow was cured in the control group [114]. These results suggest that although phage therapy holds promising applications to increase the survivability of livestock, its efficacy upon animal application should be improved. Phage-based therapy is also a promising technology to prevent the transmission of pathogenic diseases from animal-derived food to humans. Several studies have focused on phage-based biocontrol in pre-harvest procedures to eliminate harmful foodborne pathogens [115,116,117]. Niu et al. showed that administering phages infecting the harmful *E. coli* O157:H7 in feedlot cattle resulted in 16.9% of culture-negative *E. coli* O157:H7 but positive for phages, and only 6.9% positive for both *E. coli* O157:H7 and phages, indicating that phage treatment was successful in eliminating the bacteria [118]. 

With an increased interest in phage application in the agricultural and clinical fields, finding ways to administer these phages to their targeted area of effect has been a new challenge. It has been a challenge for phages designed to enter the gut, considering the harsh pH environment of the gastrointestinal tract, which can destroy phages. One emerging sub-field in phage therapy has been engineering orally delivered encapsulated phages. In one study, the researchers encapsulated four phages (wV8, rV5, wV7, and wV11) in polymer and exposed them to acidic conditions [119]. The results showed that the encapsulated phages had 13.6% recovery, whereas a complete loss of phage activity was observed in the control group. Yet another example of an effective oral delivery system uses liposomes to encapsulate phages. Two previous studies by Colom et al. encapsulated the phages UAB_Phi20, UAB_Phi78, and UAB_Phi87 in liposomes and in alginate/CaCO_3_ to test phage efficacy against *Salmonella* in poultry [120,121]. The results showed that the encapsulated phages had higher stability and higher rates of long-term protection from *Salmonella* than the unencapsulated controls. Diverse hydrogels were confirmed to encapsulate and deliver phages effectively and stably, including alginate hydrogel, PEG (Polyethylene glycol) hydrogels, and HPMC (Hydroxypropyl Methylcellulose) hydrogel [122]. Furthermore, many researchers have proposed designing polymer-encapsulated phages that can withstand gastrointestinal conditions and deliver phages to their area of effect in a pH-dependent manner. Thus, many recent studies have begun to investigate the use of pH-sensitive encapsulation material, which can precisely trigger phage release in the gut. For example, Vinner et al. found that encapsulating *Clostridium difficile* bacteriophages in Eudragit^®^ S100 with and without alginate, a pH-responsive polymer, allowed the encapsulated phages to survive and be delivered in a simulated pH 2 gastric fluid environment for hours [123]. Their finding indicates that these encapsulated phages are good contenders for targeted delivery, as the phages will only be released in the low-pH environment of the gut and not in other parts of the gastrointestinal tract. Not only is it important for encapsulation materials to allow phages to survive in the harsh pH environments of the gut, but the phages should also be released in the exact target area for effective treatment. 

Although phage-based therapy certainly has its advantages, there are disadvantages to using this procedure in a medical setting. Excessive use of phage therapy can lead to the potential evolution of pathogenic bacteria to become phage-resistant. Thus, this means that in the future, a well-designed phage cocktail that can precisely target specific bacteria pathogens must cooperate with other intervention strategies for therapeutic benefit. Additionally, phage therapy could lead to disastrous health consequences, such as virulence against the healthy gut microbiota or off-target effects. To overcome these barriers, new studies are necessary to test the efficacy of phage-based therapy in controlling and preventing human gut-related diseases in the future. Overall, phage-based therapy is still a new and emerging field, and most studies are in vitro and animal model studies with potential future applications in human medicine.

### 6.3. Gene Editing Phages

With novel discoveries in phage-based therapy, research has begun focusing on how phages can be coupled with genetic engineering for potential clinical applications in humans and animals. In particular, there has been an interest in using phage therapy to deliver CRISPR-Cas9 systems to the gut in order to make genomic edits for the microbiome. Lam et al. used mouse models to study the effects of genetically engineered bacteriophages as vehicles for CRISPR delivery into the gut [124]. The researchers engineered the filamentous bacteriophage M13 to deliver exogenous CRISPR-Cas9 DNA to *E. coli* bacterial populations of the mouse GI tract. The results reported that using phages as a delivery mechanism for CRISPR systems could lead to induced chromosomal deletions in selected bacteria in both in vitro and in vivo settings. However, some results showcase potential drawbacks of this technique, including bacterial deletion of the CRISPR DNA to escape targeting. Similarly, Gencay et al. developed a combination of four phages engineered with tail fibers and CRISPR-Cas machinery—SNIPR001—to specifically target *E. coli*. The oral gavage of a high-dose phage cocktail (2 × 10^11^ PFU) could lead to a 4 log CFU/g reduction in *E. coli* in the mouse gut without disturbing the gut background flora and mammalian health [125]. Most of all, an investigation regarding the safety and ability of phage cocktail SNIPR001 against *E. coli* infections in humans via multiple oral administration is currently ongoing as a clinical trial in the United States. Overall, these findings provide proof of concept for future studies and suggest that manipulation of the gut microbiome is possible by using an engineered phage delivery system.

### 6.4. Fecal Viral Transplantation

Yet another important benchwork tool for phage therapy applications to the human gut is fecal viral transplantation (FVT). FVT involves screening and obtaining healthy stool samples from one individual and further transplanting the healthy gut viral populations into the colon of another individual [126]. Essentially, FVT allows the virome of the healthy individual to repopulate the gut of the sick individual. Given the connection between gut microbiota and human health by phages, this technique has been proposed as a therapeutic treatment for gastrointestinal diseases. One of the primary advantages of using FVT is to prevent and treat bacterial infections in gut environments. As described earlier, numerous pathogenic bacterial species can disrupt the healthy gut microbiome and cause dysbiosis. Thus, it is critical to combat and eliminate these pathogenic agents for restoring a healthy gut. The use of bacteriophages emerges as a possible solution. The technique of fecal viral transplants can eliminate harmful bacteria from the gut environment and also facilitate the proliferation of beneficial probiotic species. A study by Rasmussen et al. investigated how FVT could promote the growth of probiotic species *Lacticaseibacillus rhamnosus* GG (LGG) and *Akkermansia muciniphila* (AKM) in the guts of mice [127]. The study revealed that mice treated with FVT had a significantly higher abundance of naturally occurring AKM compared to controls. Currently, the applications of FVT remain tested in animal models. One of the applications is to use FVT to improve the restoration of the normal bacterial gut microbiota after antibiotic treatment. Draper et al. disrupted the mice’s gut microbiome using a combination of penicillin and streptomycin [128]. After that, the bacteriome of mice with either FVT or FVT with heat and nuclease treatment (both of which killed the phages as controls) was observed. The results indicated that the fecal viral transplanted mice had a higher degree of resemblance to the pre-antibiotic treatment bacteriome. Moreover, analysis of the gut viromes of both groups of mice demonstrated that the fecal viral transplanted mice maintained the phages used in the transplantation over time, suggesting long-term benefits. Thus, their findings suggest the role of FVT as not only primary treatment but also as a way to mitigate side effects that result from other medical treatments. 

## 7. The Challenge of Phageomes in Gut Microbiome, Perspectives, and Future Directions

The underlying interaction between the gut phageome and bacteriome in animals and humans is the identity of phages as viruses of bacteria. Phage infection opens the way to commonly observed phage–bacteria interactions, including horizontal gene transfer and predator–prey ecological dynamics. Specifically, once phages infect and enter a bacterial cell, they can transduce certain functional genes and alter the bacteria’s phenotype. Likewise, in the gut environment, phages and bacteria are constantly co-evolving to infect or defend to gain an advantage over the other. The phage–bacterial interactions cause changes in gut microbiota, including fluctuations in the phage and bacterial populations and induction of prophages, and have ongoing effects on the mammalian body. Phages can turn gut microbes pathogenic through viral transduction, causing disease in the host. On the contrary, the phages also protect the mammalian hosts by infecting and lysing pathogenic bacteria in the gut, able to activate the immune system. Therefore, phage–host interactions through phage infection need to be further explored in order to understand the gut microbiome better and for future directions.

There are challenges to studying phages in the mammalian gut microbiota. In general, some phage genomes share little homology with known phage sequences. Novel phage sequences continue to be discovered in phage research, particularly in viral metagenomics. Phages are constantly evolving, and the diversity of viral genomes also hinders how well researchers can align phages with references. Metagenomic approaches have successfully made progress to facilitate the characterization of different phages in the environment, closely associated with the gut virome and bacteriome; however, at this time, pure genomic studies cannot detail specific phage–host interactions. As genomic and sequence-based tools become more popular, researchers still require information from biological experiments. The missing pieces would create a methodological bias that does not account for phage biology and evolutionary history and low sensitivity toward small DNA yields from phages. Moreover, unlike bacteria, phages lack universal phylogenetic markers, and with current protocols, gathering large quantities of viral genetic material within metagenomic data not contaminated by bacterial sequences is often complicated. Most importantly, there is limited information on the gut phageome. Several studies indicated that most viral populations could not be assigned with taxonomy, thus keeping them unknown as viral “dark matter”. Finkbeiner et al. employed a novel “micro-mass sequencing” technique to detect the presence of both known and unknown viruses in diarrhea samples [129]. Their study shows that these detected viruses are highly interconnected to gastrointestinal health; characterizing these viral genomes can lead to a broader understanding of both causes and treatments of prevalent diseases. A few studies have investigated techniques to explore this viral dark matter [130]. For instance, Benler et al. used whole-community gut metagenomics to discover 3738 complete phage genomes from 451 different genera, thus illustrating vast amounts of unknown human gut phages [21]. In the future, new studies and techniques are necessary to facilitate exploring these complex gut phageomes.

When utilizing phages in therapy or application, the stability of phages is also a challenge because it is frequently variable across phage species and formulations, which makes it difficult for researchers to maintain phage titers through experiments and clinical trials. If not frozen or cooled down, phages will spontaneously mutate over extended periods in storage, which can impair the fitness of phages and research data. In addition, there is generally a lack of quality and safety guidelines to prepare phages, especially for therapy purposes. Although there are strict regulations for pharmaceutical products, few standards have been addressed specifically for phage research and application [131]. Also, phage research lacks a simple, fast, and high-throughput method to screen phages; furthermore, the current research approaches, including double-layer agar plates, real-time PCR, and flow cytometry, are not easily compatible with all phages while producing quick results. If phages continue to be researched for therapy, improved experimental methods, including methods in genomics, molecular biology, and microbiology, are required, and standards of pharmacokinetics and pharmacodynamics of phages also need to be investigated [132]. The optimal routes of administration, dosage, and the appropriate diseases for phage therapy are also research gaps. Moreover, the problem of phage resistance has continued to occur. If researchers and healthcare providers are to move toward phages, this remains a barrier to more effective research and application. Because of this, there can be uncertainty among researchers, patients, and consumers regarding the effectiveness of phages and the arms race between phages and their bacterial hosts [133]. As phages in the gut microbiota continue to be studied, there are still gaps to bridge in the future, and current research methods pose some challenges to learning more about viral dark matter. 

Understanding the multitude of mammalian diseases that arise from the dysbiosis of the gut microbiota remains a goal for microbiology research. Many current and future studies also focus on manipulating the gut microbiome for therapeutic benefit despite the challenges and research gaps. Since the microbiome is inherently linked to various immune and metabolic responses, manipulating the microbiome by changing its composition and diversity can hold the key to treatments for different health conditions. One of the emerging techniques, known as fecal microbiota transplantations (FMTs), is to use gut microbiome for therapeutic benefit, such as IBD clinical trials; however, it is not always effective. Therefore, the transplantation of VLPs is better than the traditional FMT with several advantages. First, unlike FMT, FVT does not pose the risk of disseminating pathogenic bacterial species from a donor to the recipient because no bacteria are transferred during this procedure. Secondly, FVT could theoretically lead to higher therapeutic efficacy than FMT since researchers can carefully select and screen for particular phages or viruses from fecal samples. On the contrary, there is room to improve the FVT. Future research should focus on improving the selection and isolation of phages from fecal samples to enhance efficacious results. Additionally, human trials are required to see whether these technologies can be safely applied to clinical treatment (Figure 3).

Due to the function of fecal viral transplants in facilitating or eliminating bacterial species in the gut microbiome, an emerging field of study focuses on genetically manipulating the microbiome to help combat diseases. One of the most interesting contemporary examples of this has been using bacteriophages as vehicles to deliver CRISPR-Cas9 systems to the gut as previously mentioned. Additional research on CRISPR and phages can potentially help revolutionize this field. Several concerns regarding CRISPR and gene editing must be addressed in future research for safety purposes. For example, one primary topic in current and future studies is to reduce the off-target effects of the CRISPR-Cas9 system. Improving the specificity and screening of bacteriophages to be used as vehicles for the CRISPR systems is another potential topic for future research.

Overall, recent studies have revealed the role of phageomes in the gut microbiota, but several mysteries remain to be discovered. Additionally, there are many potential directions to expand this field for future research, including establishing core gut viral and genetic databases, studying the evolutionary adaptations of the gut microbiota to changing viral environments, and observing specific changes in host phenotypes in response to alterations in the gut phageome. Future research will hopefully discover the underlying mechanism and interactions of gut microbiota and find more efficient alternatives toward these challenges related to mammal health and diseases.

## Figures and Tables

**Figure 1 microorganisms-11-02454-f001:**
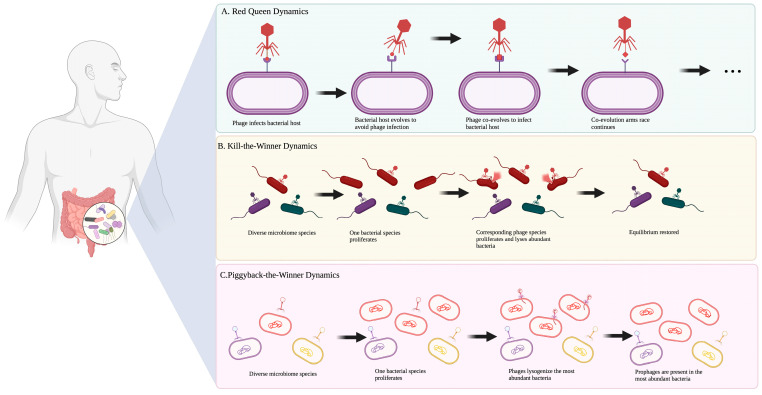
Phage–bacterial dynamics within GI environment gut. (**A**) Red Queen dynamics: a continual co-evolution between bacteria and phage to defend and counter-defend, each species running to keep up with the other; (**B**) kill-the-winner dynamics: once an individual bacteria species starts to dominate the overall bacterial population, phages lyse these cells to prevent bacterial dominance; (**C**) piggyback-the-winner dynamics: at high host bacteria abundances, phages switch to the lysogenic lifecycle to prevent a closely related phage from infecting the same bacterial cell by superinfection exclusion [51].

**Figure 2 microorganisms-11-02454-f002:**
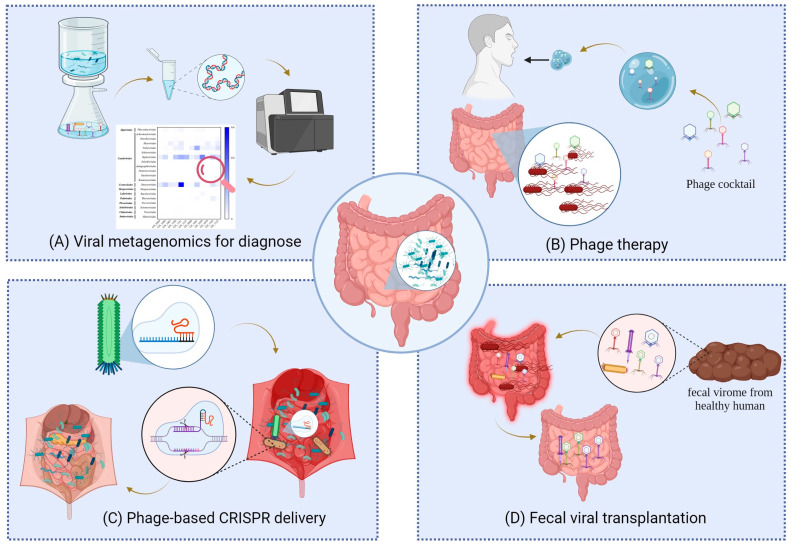
Phage-based diagnostics and therapy. (**A**) Viral metagenomics has been used to diagnose the causes of disease by determining the changes in gut virome and potentially to trace back the source of these pathogens. (**B**) Phage therapy has become a promising technique to selectively eliminate the target harmful bacteria and cure gastrointestinal diseases, such as oral administration of the phage cocktail. (**C**) Phage-based CRISPR delivery uses engineered bacteriophages as vehicles for CRISPR delivery into the gut to induce chromosomal deletions in selected bacterial pathogens in the gut microbiome. (**D**) Fecal viral transplantation repopulates the gut microbiome of the sick individual by transferring the virome from the healthy individual to eliminate harmful bacteria and enhance beneficial probiotic species within the gut microbiome.

**Figure 3 microorganisms-11-02454-f003:**
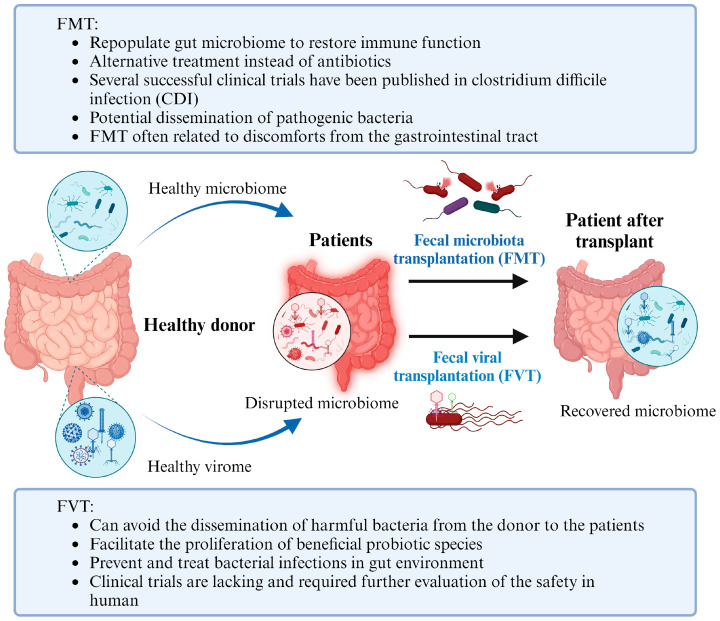
Comparison between fecal microbiota transplantation (FMT) and fecal viral transplantation (FVT). FMT can re-populate gut microbiome to recover disrupted microbiome and restore immune function via microbial interactions. FVP can reprogram the disrupted gut microbiome of patients and enhance beneficial microbial species by the interactions of viruses with surrounding microbiota within the gut microbiome. The pros and cons of FMT and FVT are included in the figure.

## Data Availability

Not applicable.

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
