# Peer review of "Gut Phageome—An Insight into the Role and Impact of Gut Microbiome and Their Correlation with Mammal Health and Diseases"

_microorganisms, 2023, doi:10.3390/microorganisms11102454_

Round 1

Reviewer 1 Report

The novelity of manuscript is underquestion. English language has good quality. Figures and tables have acceptable quality.

1. Please explain what is the difference between your manuscript and two manuscript below:

1. The human gut phageome: composition, development, and alterations in disease

Yingying Zhang, Ran Wang

Frontiers in Microbiology 14, 2023

2. The human gut phageome: origins and roles in the human gut microbiome

Eleanor M Townsend, Lucy Kelly, George Muscatt, Joshua D Box, Nicole Hargraves, Daniel Lilley, Eleanor Jameson

Frontiers in cellular and infection

microbiology, 498, 2021

2. Please check and adjust the "Reference list" based on the regulations of reference list of journal. (Titles, doi, the name of journal and ... )

Author Response

  1. Please explain what is the difference between your manuscript and two manuscript below:
  2. The human gut phageome: composition, development, and alterations in disease

Yingying Zhang, Ran Wang

Frontiers in Microbiology 14, 2023

  1. The human gut phageome: origins and roles in the human gut microbiome

Eleanor M Townsend, Lucy Kelly, George Muscatt, Joshua D Box, Nicole Hargraves, Daniel Lilley, Eleanor Jameson

Frontiers in cellular and infection microbiology, 498, 2021

Response: We appreciate the reviewer's comment. Here are several differences between our review article and the two references above:

  1. Our review article focused on mammal animals, including humans, animal models (i.e., mice), and farm animals (i.e., pigs, chicken), for clinical and agricultural purposes. The two references mainly focused on the human gut phageome.
  2. The first reference reviewed the taxonomic composition and spatial distribution of phageome in the human gut, specifically focusing on the major common viral families, such as Caduovirales, Microviridae, and Crassphage. In our review article, we highlighted an overall profile of diverse phages across the gastrointestinal tract and their dynamic roles in shaping the surrounding microorganisms.
  3. The second reference provided a timeline regarding the developed methods to identify and characterize phages in the human gut. However, the current technologies mentioned in our research were more focused on phage-based diagnosis, therapy, and biocontrol.
  4. Both reference articles summarized the role of phage in human diseases. In our review article, in addition to the latest research updates on the role of gut phageome in the association of mammalian health and diseases, we also provide potential solutions for disease treatment and prevention.

  1. Please check and adjust the "Reference list" based on the regulations of reference list of journal. (Titles, doi, the name of journal and ... )

Response: We appreciate the reviewer's comment and have updated the reference list based on the journal requirements.

Reviewer 2 Report

The article is well-written, and the use of the English language is of high quality. Your research on the gut phageome and its various applications is undoubtedly valuable and contributes to the field.

I have a few suggestions that I believe could enhance the clarity and impact of your article:

1. Concluding Sentence for Abstract: Consider adding a concluding sentence to the abstract that summarizes the overall significance of your research or its potential implications for the clinical and agricultural fields.

2. Definitions for Key Terminology: While your introduction effectively introduces key terminology such as "gut phageome" and "lysogenic phages," it might be helpful to provide brief definitions or explanations for these terms.

3. Expanded Examples: The examples you provided about phage capsid protein mutations and tail fiber genes are excellent in illustrating the adaptability of phages. To further enhance reader understanding, you could consider expanding on these examples to provide more detail or context.

4. Discussion of Limitations and Challenges: It would be beneficial to discuss potential limitations or challenges associated with viral metagenomics and phage therapy. Providing a balanced view by addressing these aspects can strengthen your article.

5. Recent Studies and Breakthroughs: Mention any recent studies or breakthroughs related to gene editing with phages to demonstrate the current state of research in this area. This can help readers understand the context and relevance of your work.

6. Clinical Trials on Fecal Viral Transplantation: If possible, include information about any ongoing or planned clinical trials related to fecal viral transplantation. Additionally, consider incorporating a flowchart that outlines the stages of such trials. This would provide valuable context for your readers.

7. Safety Considerations for FVT: Given the increasing interest in fecal viral transplantation (FVT), it would be advisable to discuss safety considerations related to FVT, especially in comparison to traditional fecal microbiota transplantation (FMT). Creating a table or figure that compares the advantages and disadvantages of FVT and FMT could be particularly insightful.

8. After the initial introduction of microorganism in the text, you can simply refer to it using the abbreviated form.

I believe that implementing these suggestions will further enhance the quality and impact of your article. Your research has the potential to contribute significantly to our understanding of the gut phageome and its applications.

Author Response

  1. Concluding Sentence for Abstract: Consider adding a concluding sentence to the abstract that summarizes the overall significance of your research or its potential implications for the clinical and agricultural fields.

Response: We appreciate the reviewer's comment and have updated the abstract in the revised manuscript (line 21-24).

  1. Definitions for Key Terminology: While your introduction effectively introduces key terminology such as "gut phageome" and "lysogenic phages," it might be helpful to provide brief definitions or explanations for these terms.

Response: We appreciate the reviewer's comment. The definitions of key terminology were provided in the introductions section (line 60-69: Lytic phage, lysogenic phage, gut phageome).

  1. Expanded Examples: The examples you provided about phage capsid protein mutations and tail fiber genes are excellent in illustrating the adaptability of phages. To further enhance reader understanding, you could consider expanding on these examples to provide more detail or context.

Response: We appreciate the reviewer's comment and have updated the details in the revised manuscript (line 129-144).

  1. Discussion of Limitations and Challenges: It would be beneficial to discuss potential limitations or challenges associated with viral metagenomics and phage therapy. Providing a balanced view by addressing these aspects can strengthen your article.

Response: We appreciate the reviewer's comment. The limitations and challenges of viral metagenomics and phage therapy have been updated in the revised manuscript (line 366-373, line 460-470, line 539-561, line 562-584).

  1. Recent Studies and Breakthroughs: Mention any recent studies or breakthroughs related to gene editing with phages to demonstrate the current state of research in this area. This can help readers understand the context and relevance of your work.

Response: We appreciate the reviewer's comment and have updated the latest research in the revised manuscript (line 483-489).

  1. Clinical Trials on Fecal Viral Transplantation: If possible, include information about any ongoing or planned clinical trials related to fecal viral transplantation. Additionally, consider incorporating a flowchart that outlines the stages of such trials. This would provide valuable context for your readers.

Response: We appreciate the reviewer's comment. We searched NCBI clinicaltrials.gov, which collected information of clinical trials worldwide (https://www.clinicaltrials.gov/). However, no clinical trials regarding FVT was found. However, we have added a new Figure 3 for the current status of FVT (con).

  1. Safety Considerations for FVT: Given the increasing interest in fecal viral transplantation (FVT), it would be advisable to discuss safety considerations related to FVT, especially in comparison to traditional fecal microbiota transplantation (FMT). Creating a table or figure that compares the advantages and disadvantages of FVT and FMT could be particularly insightful.

Response: We appreciate the reviewer's comment and have added a new figire, Figure 3 in the revised manuscript.

  1. After the initial introduction of microorganism in the text, you can simply refer to it using the abbreviated form.

Response: We appreciate the reviewer's comment and have used abbreviated names of the microorganisms in the revised manuscript.  

I believe that implementing these suggestions will further enhance the quality and impact of your article. Your research has the potential to contribute significantly to our understanding of the gut phageome and its applications.

We thank the reviewers' comments. 

Round 2

Reviewer 1 Report

No more comment.